# Using the National Early Warning Score (NEWS) outside acute hospital settings: a qualitative study of staff experiences in the West of England

Emer Brangan,[1,2] Jonathan Banks,[1,2] Heather Brant,[1,2] Anne Pullyblank,[3,4] Hein Le Roux,[3,5] Sabi Redwood[1,2]

EB and JB are joint first authors.

For numbered affiliations see end of article.

**Correspondence to**
Dr Emer Brangan;
e.brangan@bristol.ac.uk

## ABSTRACT

**Objectives** Early warning scores were developed to improve recognition of clinical deterioration in acute hospital settings. In England, the National Early Warning Score (NEWS) is increasingly being recommended at a national level for use outside such settings. In 2015, the West of England Academic Health Science Network supported the roll-out of NEWS across a range of non-acute-hospital healthcare sectors. Research on the use of NEWS outside acute hospitals is limited. The objective of this study was to explore staff experiences of using NEWS in these new settings.

**Design** Thematic analysis of qualitative semi-structured interviews with purposefully sampled healthcare staff.

**Setting** West of England healthcare settings where NEWS was being used outside acute hospitals—primary care, ambulance, referral management, community and mental health services.

**Participants** Twenty-five healthcare staff interviewed from primary care (9), ambulance (3), referral management/acute interface (5), community (4) and mental health services (3), and service commissioning (1).

**Results** Participants reported that NEWS could support clinical decision-making around escalation of care, and provide a clear means of communicating clinical acuity between clinicians and across different healthcare organisations. Challenges with implementing NEWS varied—in primary care, clinicians had to select patients for NEWS and adopt different methods of clinical assessment, whereas for paramedics it fitted well with usual clinical practice and was used for all patients. In community services and mental health, modifications were 'needed' to make the tool relevant to some patient populations.

**Conclusions** This study demonstrated that while NEWS can work for staff outside acute hospital settings, the potential for routine clinical practice to accommodate NEWS in such settings varied. A tailored approach to implementation in different settings, incorporating guidance supported by further research on the use of NEWS with specific patient groups in community settings, may be beneficial, and enhance staff confidence in the tool.

### Strengths and limitations of this study

► This study represents new research on use of the National Early Warning Score (NEWS) in prehospital, primary and community healthcare settings.
► In-depth qualitative interviews with healthcare professionals facilitate understanding of barriers and facilitators to using NEWS in prehospital, primary care and community settings.
► Interviewees were recruited from a wide range of healthcare organisations.
► There were limited numbers of interviewees from each healthcare organisation.
► The study was restricted to one region of England, UK, which at the time of the research was the only region where NEWS had been rolled out across the healthcare system.

## INTRODUCTION

Early warning scores (EWS), or physiological 'track and trigger' systems, are designed to support healthcare professionals (HCPs) to identify and respond to acutely unwell patients at risk of clinical deterioration. EWS are established in UK acute hospital care[1 2]; however, different systems have been used in the UK National Health Service (NHS), and concerns about a lack of standardisation[3 4] prompted the Royal College of Physicians (RCP) to develop the National Early Warning Score (hereafter referred to as 'NEWS') in 2012. They recommended adoption of NEWS across the NHS to ensure a standardised tool that would be interpreted consistently by HCPs.[4] NEWS uses six physiological measurements: respiratory rate, oxygen saturation, temperature, systolic blood pressure, heart rate and level of consciousness; each scored 0–3 and added together for an overall score, with higher scores indicating the need for more intensive monitoring and/or clinical intervention. An

additional two points are added if the patient is receiving oxygen therapy.[4]

While the RCP report focused on using NEWS in acute hospital care, they also proposed using the tool in other healthcare settings including primary care, community hospitals and ambulance services.[4] More recently, NHS England endorsed the use of NEWS by ambulance services, mental health hospitals and prisons, and also encouraged further evaluation in primary care.[5]

In 2015, the West of England Academic Health Sciences Network (WEAHSN)[6] identified NEWS as a key component of its patient safety programme[7] and promoted the tool across its footprint (West of England).[8] This included healthcare sectors outside acute hospitals, such as primary care and ambulance services, where it was advocated for the assessment of acutely unwell patients and handover of care between healthcare settings, organisations and HCPs.

However, research on the use of EWS, including NEWS, outside hospital settings is very limited, and has focused primarily on evaluating the predictive accuracy of the tool in these settings.[9] Effective use of NEWS is also reliant on staff engagement with the tool[10] and it is equally important to understand the views and experiences of the staff who use NEWS, particularly when it is being used in a new way. To date, qualitative research outside the acute hospital sector has been limited to a small scale study of NEWS in a regional ambulance service[11] which is unable to offer insight into the use of the tool in other sectors such as primary care.

We undertook a qualitative study of use of NEWS by HCPs outside the acute hospital sector in the West of England following its introduction and promotion by the WEAHSN, and included participants from: primary care; ambulance services; acute advice and referral management services; community nursing and rapid response teams; and mental health services. Our focus was on how staff used NEWS, their views on the role of the tool in assessing acute illness and its role in the escalation of care.

## METHODS

This was a qualitative semi-structured interview study with HCPs in the West of England who either used NEWS or were involved in its implementation outside the acute hospital sector. Recruitment commenced using purposive sampling in collaboration with WEAHSN to identify and contact individuals from all relevant sectors and organisations involved in the regional roll-out of NEWS with details about the research. Thereafter, snowball sampling was used with the objective of obtaining a diverse sample regarding sector, organisation, professional role, role in relation to NEWS and orientation to NEWS (both positive and critical views were actively sought). Information about the study was sent to potential participants by email, with an invitation to contact the research team if they were interested in participating in an interview.

Interviews were carried out by EB and JB by telephone or face to face according to participant preference. A topic guide (see online supplementary file) was used to focus the interviews, while allowing participants to raise topics not covered by the guide. This guide was informed by a scoping review of relevant literature by JB and suggestions from our multiprofessional study team, and modified as data analysis progressed.

With informed consent, interviews were audio recorded, transcribed verbatim, anonymised and imported into NVivo 10 (QSR International). Transcripts were analysed thematically[12] using a data-driven inductive approach to identify patterns and themes of particular salience for participants and across the dataset. Analysis began alongside data collection, with ideas from early analysis informing later data collection.[13]

Analysis of individual transcripts commenced with open coding—EB and HB each coded a sample of early transcripts and jointly developed an initial coding framework, which was added to and refined as new data were gathered. A sample of transcripts were double coded independently by JB to ensure robust analysis. At each stage, any coding differences were discussed—these were minor and were resolved by clarifying and agreeing descriptions for the relevant codes. The coding framework was applied to all transcripts by EB and HB who thereafter developed broader categories through comparison across transcripts, and higher level recurring themes were identified. Two participants provided written feedback on the summarised findings. Members of the study team met regularly to discuss emerging themes.

### Patient involvement

An established public involvement panel met twice to discuss anonymised data extracts and emerging themes, and the panel's feedback was used to inform development of the coding framework.

### RESULTS

Twenty-five staff employed by 15 organisations working in the West of England were interviewed between December 2016 and May 2017. Interviews had an average duration of 32 min, with 15 participants choosing to be interviewed by telephone. Table 1 provides details of participants' backgrounds.

Participants had a broad range of clinical experience, from a general practitioner (GP) who had been qualified for less than 2 years, to nurses who had been practising for several decades. Most participants had several years of experience relevant to their current role. The majority of participants reported 2 or more years of experience with NEWS outside the acute hospital sector. However, this was dependent on when participants' organisations had introduced NEWS. All participants reported at least 7 months experience of working with NEWS. There was also variation in use in relation to clinical practice from

**Table 1** Interview participants' professional backgrounds

| Sector/professional background | Participants |
|---|---|
| Primary care general practitioner | 7 |
| Primary care nurse | 2 |
| Community services nurse | 4 |
| Mental health nurse | 2 |
| Mental health clinical trainer | 1 |
| Ambulance trust paramedic | 3 |
| Commissioner | 1 |
| Acute interface doctor | 2 |
| Acute interface nurse | 1 |
| Acute interface call handler | 2 |
| *Total participants* | *25* |
| Notes on sectors | |
| Primary care | Participants from general practice and out of hours primary care services. |
| Community services | Participants providing/leading nursing services delivered in patients' homes, including nursing care for housebound people with long-term conditions, or more intensive shorter term care for those who were acutely unwell (rapid response/urgent care teams), with a view to supporting people at home and avoiding hospital admissions. |
| Acute hospital care interface | Participants working in services at the interface between primary/community care and acute hospitals, including acute advice and/or referral management services, and hospital admitting departments. |

*Anonymisation:* Quotes are labelled using letter codes allocated to each participant.

regular (eg, paramedic) to infrequent (some primary care staff).

We developed four themes from the data which highlight the ways that NEWS was used across different sectors, how it affected how HCPs work with patients and each other and how it could both support and challenge usual clinical practice outside of acute hospital settings. These were: NEWS and communication; NEWS in prioritisation of care; NEWS and clinical judgement; and integrating NEWS into clinical practice.

## NEWS and communication

Part of the rationale of NEWS is to provide a simple and standardised way of conveying a patient's clinical acuity, using an aggregated score, rather than relying on narrative description combined with a variable selection of physiological observations. In this way, NEWS could function as a common language in communication between services, and our participants described the tool having the effect of speeding up interactions and responses by succinctly communicating a HCP's basis for concern:

> I think it's difficult to convey a patient's condition over the phone and sometimes in the past I have been saying, 'They have got tachycardia and they don't look well.' If you say, 'Actually they have got a NEWS score of six.' Suddenly they say, 'Well I think we better see them.' (Nurse Q: primary care)

NEWS could give HCPs leverage in escalating care, particularly when staff were not known to each other or there was a perceived clinical status gap. Several participants reported that having NEWS increased their confidence to communicate their concerns in such circumstances:

> One of the nurses saw somebody with a NEWS score of seven. She thought he was septic…. She said that before, she'd have had to speak to one of us before she got the approval for doing the admission. But because he had got a NEWS of seven, she was able to phone the ambulance service, phone the ED, and get it sorted out. (GP Y)

In some instances, NEWS was being used by HCPs as a strategic tool to negotiate a patient's admission to hospital, rather than as a clinical tool to help determine whether admission was warranted:

> I guess I would use it if I'm being asked for it when I admit people and it makes my life easier to admit people, then I'd be more likely to use it…it's one less thing to have hassle from an admitting person about if you've done it. (GP L)

However, there were differences across organisations in how effective NEWS was in communication. Different degrees of recognition or understanding of NEWS was

a factor—NEWS could not help communication if one party to that communication did not use/understand the tool:

> You have to put in the word 'sepsis', I think, otherwise they have no idea what a NEWS… We're told that the ambulance crews are using it, but I don't think that's across the board… And I don't think the GPs have any idea what we're talking about. (Nurse T: community services)

Communication was also problematic if one party did not perceive NEWS to be of value, or relevant to that discussion:

> There is a sort of 'just take the referral and I've done the assessment and that is all you need to know, don't ask me to be doing resps and temperature and blood pressure when it's not relevant', in their [some GPs'] eyes. (Nurse I: referral management/acute advice)

### NEWS in prioritisation of care

NEWS was also intended to support triage and prioritisation of care. Participants described using the tool to help make decisions regarding when a patient should be seen and by whom—it was taken into account when making and accepting referrals, or specifying ambulance response times. A rising NEWS score was seen as a particularly clear indicator that a patient needed to be prioritised:

> Although emergency departments understand if a blood pressure or heart rate is deteriorating, [NEWS] encapsulates all those baseline observations, so that we know that, actually, somebody who, originally, 20 min ago, had a NEWS score of four, now has a NEWS score of seven, and this is clearly identifiable as somebody who may need to go to the front of the queue (Paramedic P)

Participants also saw NEWS as providing an objective justification for referral decisions which might be challenged—conferring greater confidence in making decisions both to refer (higher NEWS) or not to refer (low NEWS):

> Making the decision not to admit someone or not to refer someone, which we have to do most of the time, there's potentially a lot of comeback on you for not doing that, so anything that covers you and helps protect you or back up your decision making is potentially useful. (GP L)

However, some participants voiced concerns regarding the potential for NEWS to be used inappropriately to deprioritise patients whose acuity was not reflected by their score—head injury, stroke and some acute cardiac complaints were given as examples. In the case below, of two patients with sepsis, a GP questioned whether the patient with a higher NEWS score had been at greater risk and highlighted the lower scoring patient's history and comorbidities:

> [He] Just looked unwell from the minute he came in. Breathing very fast, again. Very high fever, with quite nasty coronary disease; he was waiting to have an angiogram next week. So, an elderly, frail sort of gentleman. NEWS score of five…. I was thinking, 'He's no less unwell than the other chap.' [who had sepsis with a NEWS of seven] He was in exactly the same sort of situation, really, and in fact, he's got known coronary disease for which he's awaiting an intervention. In a way, maybe he's more at risk, because sepsis could precipitate something like a coronary event in someone like him. (GP N)

Participants emphasised the need for NEWS to be used alongside other sources of information—such as history and clinical judgement—when making decisions regarding escalation of care, and never as a 'rule out':

> The problem with the acute work that we do is there's an admission avoidance remit, but I don't think NEWS should ever be used as a tool to stratify patients who perhaps don't need to come to hospital … I think it's a rule in rather than a rule out. (Doctor G: referral management/acute advice)

### NEWS and clinical judgement

Participants regularly described the tool as an aid to clinical assessment, to be used alongside clinical judgement:

> It is a clinical tool to aid your clinical decision-making. We very much advocate and stress to staff that it's not there to replace your clinical judgement… they don't just go off the NEWS score. It's just a tool to help. (Paramedic P)

NEWS often aligned well with and supported a HCP's impression of a patient's clinical status based on other information:

> When he came in …some sixth sense in me thought he was more unwell than he seemed. I did a NEWS score on him, and I think it was six or seven … it enabled me to feel confident that my gut reaction that this guy was sick was turned into something, again, more objective. (GP V)

There were also instances where NEWS provided a helpful challenge to clinical judgement:

> I think I have had like sepsis patients who looked deceivingly well kind of outwardly and before we may have just kind of sort of 'oh they are slightly tachycardic, but they are fully alert and talking to me quite normally' … and then put it all together and their NEWS score is really really high that it's kind of alarmed you, that actually they are quite unwell. (Paramedic J)

However, a score which was not well aligned with a HCP's clinical judgement of the risk to an individual patient might also be 'overruled':

The high NEWS score isn't necessarily something that you would automatically admit. So, the young-ish person perhaps with a nasty case of the flu, their heart rate can run at 120, their temperature can be 38, 39 but actually they've got enough physiological reserve that that's not going to be a problem for them and a bit of paracetamol, some fluids and some care with safety netting is enough, so, I think it's taking it in context (Doctor G: referral management/acute advice)

Some HCPs found conflicts between NEWS and their clinical judgement uncomfortable. A nurse working in primary care perceived nurses as more vulnerable to censure than doctors if they 'overruled' a non-zero NEWS based on their clinical judgement. This participant described conflicts between NEWS and clinical judgement as causing anxiety, and reported responding by seeking a second opinion, or safety netting:

…Because I recognise that kind of anxiety situation, informally I've been asking them to be reviewed by the duty doctor. But that's not always possible because sometimes I'm on my own…In that situation I really just do have to man up and make a decision…That is difficult. It's judgement, but you have to do it in the end. (Nurse Q: primary care)

### Integrating NEWS into clinical practice

With early warning scores traditionally more commonly used in acute hospital settings, outside of these settings HCPs needed to find ways to incorporate NEWS into their clinical practice. For paramedics, NEWS fitted well with usual clinical practice:

As far as clinicians go, paramedics have been quite good at taking robust sets of observations for quite a long time…NEWS fits that. So, there wasn't a change in practice for us. (Paramedic O)

Participants consistently identified GPs as the group least likely to be familiar with or receptive to NEWS. Some participants attributed this to NEWS being challenging to usual clinical practice for GPs:

The problem with GPs is time. Two minutes is quite a significant time. Our patients always have clothes on them, so you have to get them undressed … We have 10 min appointments. The nurses have 20 min, half an hour. In hospital, somebody else does the observations (GP Y)

A further issue was different diagnostic cultures. While paramedics and some nurses were seen as routinely taking full sets of observations of vital signs, GPs were described—by themselves and other HCPs—as more reliant on history, symptoms and clinical instincts, with physiological observations used selectively. Unlike the ambulance service, where NEWS was used with every patient, participants referred to the need to make judgements about

who to use NEWS with in primary care. A GP highlighted the low proportion of patients presenting to primary care who warranted NEWS and the risk of medicalising through overuse:

Who are you going to use it for? We see so many people with self-limited things, where actually, what you're really thinking is, 'I don't want to medicalise this too much.' …These are people who I can tell, by eyeballing them, they look pretty well, and from talking to them, they're pretty well. … And if every time someone comes in with a bit of a sore throat, and you end up doing blood pressure, temperature and all the rest of it on them, then they're going to think, 'Oh, gosh. It's a good thing I came. I'll need to come next time I've got a sore throat.' (GP N)

Participants working in community and mental health services reported an increased focus on vital signs driven by NEWS and regarded this as a positive change. Those working in mental health noted that staff's experience in assessing physical health was often more limited, which could present challenges when measuring NEWS. However, it also meant that the clinical guidance associated with different NEWS scores was seen as particularly helpful.

HCPs from community and mental health settings described NEWS being used routinely with all patients in some services, including when those patients were not acutely ill. Local protocols for clinical responses to NEWS were developed to take account of organisational constraints, such as not being set up for regular patient monitoring, or not having easily accessible resources for escalation. Some services also adapted NEWS protocols to account for specific patient groups being cared for:

There was quite a lot of anxiety around introducing NEWS with eating disorders…and my understanding is, particularly with things like anorexia, actually they will always have, for example, a low temperature and actually if their temperature was to go within the normal range that would indicate actually they are pyrexial and a duty Doctor isn't going to understand that necessarily. So there was a risk with that… So they'd reset the triggers in that service (Nurse C: mental health)

When patients were not acutely ill, NEWS was used at scheduled intervals ranging from daily (older adult inpatient wards in a mental health service) to 3 monthly (a community nursing service). Thus, in some cases, NEWS was being used over radically different timescales to those in acute hospital settings. The rationale was that such scores provided individual baselines which contextualised episodes of acute illness, and also enabled detection of subtle and longer term deterioration. NEWS was, effectively, 'repurposed' in these settings.

HCPs expressed a desire for more evidence to underpin the use of NEWS in their own context. There was much

interest in the incidence and significance of different NEWS scores in various populations—for example, those with chronic obstructive pulmonary disease (COPD) or other chronic conditions—and whether the same triggers and clinical responses were appropriate as for an acute hospital inpatient population.

## DISCUSSION

Our study looked at the experience of HCPs using NEWS outside acute hospital settings, a use for which EWS were not originally designed. Participants described a number of benefits of using the tool: it provided a structured and objective way of communicating the severity of a patient's clinical condition, information which can be difficult to convey narratively, especially between members of different organisations who may have different clinical training and skills; it facilitated communication across hierarchical boundaries—the NEWS score had meaning regardless of a HCP's grade; NEWS could work as a clinical check for HCPs in confirming, and sometimes challenging, their clinical judgement; and NEWS could help in making decisions about how to prioritise care.

There were also challenges associated with the tool. Some participants questioned NEWS' role in prioritisation of care, and tension between NEWS and clinical judgement created uncertainty and discomfort in some cases. The RCP recommendations stated that NEWS should be used as an aid to clinical assessment, not as a substitute for clinical judgement, and that concern about a patient's clinical condition should always override NEWS.[4] The importance of using NEWS in this way was emphasised by several participants, and some remained concerned that the tool might be used inappropriately.

Some HCPs raised concerns about adopting a tool which they saw as having been designed for, and validated in, a very different context. Incorporating NEWS into clinical practice varied between organisations. For ambulance staff, the tool integrated smoothly into their workflow. However, in primary care, NEWS was only relevant to a small proportion of the patients seen, so HCPs faced the challenge of patient selection as well as not routinely working with all the physiological measures required for NEWS. In community services and mental health organisations, the tool was used in an adaptive way, sometimes beyond its original design, and with a perceived need for modifications to make it relevant to some patient populations. NEWS2, an updated version of the tool published by the RCP in December 2017,[14] incorporates changes intended to support use of the tool with patients with COPD, addressing one specific concern raised by participants in our study.

To our knowledge, this is the first in-depth qualitative study to examine the use of NEWS outside the acute hospital sector. In the West of England, NEWS has been introduced to a wide range of HCPs and organisations which increases the challenges associated with implementation and makes qualitative insight particularly valuable for policy-makers. The study is timely as NHS England recently endorsed the use of NEWS by ambulance services, mental health hospitals and prisons. They, along with the RCP, also recommended further evaluation in primary care.[5 14] By focusing our research on a wide range of organisations and HCPs, we were able look at experiences of using the tool within different organisations as well as in interactions between them, particularly in relation to communication during referrals. However, this also represents a weakness as we could only interview a small number of participants from each sector. The organisations in our study had implemented NEWS at different time points which means that some of the findings about communication being limited by HCPs not using it in some areas could be related to different stages of take-up in the various organisations. This differential take-up also influenced the amount of experience different participants had of using NEWS, which may have affected their views.

There is limited research on the use of NEWS outside the acute hospital sector. Several quantitative studies investigated the sensitivity and specificity of the tool but these have primarily been in the paramedic profession and ambulance sector.[15–19] There has only been one small-scale qualitative study but again this was restricted to the ambulance service.[11] The other major point of comparison is with the introduction of EWS into the acute hospital sector as this gives insight into the way that HCPs have reacted to and incorporated the tool into their clinical work.

Hospital-based research has shown that EWS and similar tools can facilitate improved communication between nurses and doctors,[20–22] securing attention because they provide a precise language for communicating concern[23–25] and can facilitate increased confidence in using medical language.[25] These studies resonate with our results and the potential for improved communication facilitated by NEWS, particularly the impact of the tool on communication across professional boundaries and hierarchical strata. However, EWS tools have also been shown to be a source of both interprofessional and hierarchical tension in the hospital sector—in some cases, their use was seen by senior colleagues, or other professionals, as indicating poor medical knowledge and skills.[20 26] Overall, we did not see substantive evidence for this in our data, but there were indications of some tensions between primary care and other sectors around NEWS that were related to the reluctance of some primary care clinicians to use the tool. Hospital-based research indicated that nurses found EWS tools supported decision-making and made clinical practice more consistent and reliable,[20] particularly for junior and less experienced staff.[22] However, research also indicated that experienced staff were more adept at using EWS tools as an adjunct to clinical judgement whereas junior staff were more likely

to see scores as the primary source of clinical decision-making.[27] While our data do not show such a clear relationship, there was a concern that the tool could come to dominate decision-making at the expense of clinical judgement and that it should not be the sole arbitrator for prioritising care.

This study demonstrated that NEWS can work for staff outside acute hospital settings to support clinical decision-making, communication and escalation of care. The tool's potential contribution to communication is particularly important in the community because of the different organisations that may be involved in the care and referral of patients whose clinical condition is deteriorating.

In the acute hospital sector, NEWS is used primarily by nurses, but using it outside this sector necessitates involvement from a much wider group of clinicians, and this presents more of a challenge to adoption. We have highlighted the challenges in using NEWS in primary care and other community settings. It may be the case that for further implementation initiatives a more tailored approach to the different healthcare settings could be developed.[10] Such an approach could include developing more practical guidelines about patient selection; how to incorporate use of the tool with clinical judgement and specific patient populations; defining appropriate triggers and associated actions for each setting, and using the score alongside these. Research investigating the distribution of NEWS scores in different patient populations outside the acute hospital sector[28] would be helpful in both developing such guidance and enhancing HCPs' confidence in the tool.

**Author affiliations**
[1]The National Institute for Health Research Collaboration for Leadership in Applied Health Research and Care West, University Hospitals Bristol NHS Foundation Trust, Bristol, UK
[2]Population Health Sciences, University of Bristol Medical School, Bristol, UK
[3]Patient Safety Collaborative, West of England Academic Health Science Network, Bristol, UK
[4]General Surgery, North Bristol NHS Trust, Bristol, UK
[5]Governing Body, NHS Gloucestershire Clinical Commissioning Group, Brockworth, UK

**Acknowledgements** The authors thank the participants, and their employing organisations, for their generosity with their time. Thanks to Joanna Garrett of the West of England Academic Health Science Network (WEAHSN), and Penny Whiting of NIHR CLAHRC West, for comments on the text. We also thank Ellie Wetz, Ann Remmers and Emma Stone (WEAHSN), and Niamh Redmond and Katie Warner (NIHR CLAHRC West) for their contributions to the research project management group meetings.

**Contributors** AP, SR and JB conceived the study. EB and JB generated the data and drafted the original manuscript. EB and HB analysed the data with guidance from JB and SR. HLR, SR, AP, HB, JB and EB contributed to the interpretation of findings and revisions to the manuscript. All authors approved the final version of the manuscript.

**Funding** This work was jointly funded by the National Institute for Health Research Collaboration for Leadership in Applied Health Research and Care West at University Hospitals Bristol NHS Foundation Trust and the West of England Academic Health Science Network. NIHR CLAHRC West (University Hospitals Bristol NHS Foundation Trust) received support from the West of England Academic Health Sciences Network which part funded the work.

**Disclaimer** The views expressed are those of the author(s) and not necessarily those of NHS England, NHS Improvement, the NIHR or the Department of Health and Social Care.

**Patient consent** Not required.

**Ethics approval** Approved by the University of Bristol Faculty of Health Sciences Research Ethics Committee (Application number 38181) and the Health Research Authority (IRAS project ID 214672, protocol number 2677).

**Provenance and peer review** Not commissioned; externally peer reviewed.

**Data sharing statement** Anonymised interview transcripts are available for use by bona fide researchers subject to assessment of requests by the University of Bristol Research data service DOI: 10.5523/bris.jh9lbeb3f1ng27m9ykk3dti9p

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
