## [Reviewer comments · BMJ Open]

ARTICLE DETAILS

TITLE (PROVISIONAL)	Using the National Early Warning Score (NEWS) outside acute hospital settings: A qualitative study of staff experiences in the West of England
AUTHORS	Brangan, Emer; Banks, Jonathan; Brant, Heather; Pullyblank, Anne; Le Roux, Hein; Redwood, Sabi

VERSION 1 – REVIEW

REVIEWER	Daniel Potter University of Leicester College of Medicine Biological Sciences and Psychology
REVIEW RETURNED	11-Mar-2018

GENERAL COMMENTS	This is an interesting piece of work which is successful in its aim of exploring health care practitioner experiences of using the NEWS in a variety of different settings outside of the acute hospital. The authors acknowledge that the small sample size in this study is its main weakness. The findings in this paper form a useful start point for further quantitative studies examining staff experiences and the implementation of the NEWS outside of the acute hospital. Overall, this study merits publication.
--

REVIEWER	Erina Ghosh Philips Research North America, Cambridge, MA, United States
REVIEW RETURNED	07-May-2018

GENERAL COMMENTS	This manuscript examines the usability of NEWS in non acute care areas. The use of NEWS in acute care has been validated but in non- acute care areas the benefits are unknown. This is an important topic as successful implementation of NEWS can better standardize care & aid in patient transfer. The authors performed a qualitative study by interviewing care providers from different settings on their use of NEWS. The study also lacks clear outcome definition. Comments: Study design: While qualitative interviews are useful, what would show benefit is if the authors used structured interviews or surveys so the responses could be compared against different correspondents. The current study design makes it difficult to synthesize the learnings from the interviews & discourages reproducibility. Results: Due to the qualitative nature of the study the results are presented thematically using direct quotes. While this is interesting to read, it is difficult to grasp the key ideas conveyed by the respondents. It might be useful if the authors could analyze the transcripts and classify the positive & negative comments for each
---

	theme. A more detailed description of the customization of NEWS to different outpatient/ non- acute settings might be useful for others looking to reproduce this work. Also information on users level of clinical experience & years of experience using NEWS would be good to know. Limitations: One major limitation of the study is the relatively small sample size & geographic localization. These should be elaborated on.
--	---

REVIEWER	Jørgild Karlotte Jensen VID Specialized University, Oslo
REVIEW RETURNED	18-May-2018

GENERAL COMMENTS	Thank you for the opportunity to review this manuscript. This is a topic that can have major implications for patient care and outcomes outside acute hospital settings. Please find some feedback below to further improve your paper. Main Text: In the method section the thematically analysis is well described. I wonder was there any differences in the coding among the authors and if, how did you solve this? In the results section, another reference is used to illuminate the findings. This could be done in the discussion section to ensure that only findings are presented in the result section. In the discussion section page 9 lines from 41-47, I miss some references that you refer to as earlier research.
---

REVIEWER	Miss Candice Downey University of Leeds, United Kingdom
REVIEW RETURNED	03-Jun-2018

GENERAL COMMENTS	Thanks for this interesting paper. This work addresses a relevant clinical question using appropriate methodology. I would recommend it for publication. I have a few minor comments which, if the authors choose to adress them, would elevate the paper further:  1) Please include the statement regarding ethical approval (currently in the footnotes) in the main text. 2) Please consider providing a little more information as to how the topic guide was produced. Was a formal literature review undertaken? How much was based on a priori theories? 3) It would be helpful in your table (Results section) to understand the differences between the healthcare professional groups (perhaps years since qualification, years of experience with early warning scores). For instance, duration of experience with NEWS will be an important factor in HCP's experiences, and in how the reader interprets the results. You allude to this in the discussion but I think a demographic table in the Results section would be helpful.
--

VERSION 1 – AUTHOR RESPONSE

Reviewer 1 comments

Thank you for your comments and we agree that further research, including quantitative studies, on use of NEWS outside of acute hospitals would be beneficial. This study formed part of a wider evaluation and papers have been submitted in relation to systematic reviews of the predictive

accuracy of early warning scores both within and outside hospital settings. There has also been a quantitative study looking at NEWS scores at referral alongside clinical outcomes. These papers are currently in submission and/or under review and as such it is not possible to cite them at this stage.

Reviewer 2 comments

Below we respond to your specific comments (which we quote for clarity regarding which aspects each response relates to). As several of the comments appear to relate to quantitative versus qualitative study designs, and appropriate measures of quality / best practice for different methods, we hope that the changes which we have made to our 'Strengths and limitations section' in response to editorial comments will help provide clarity regarding the reasoning behind our choice of study design and methods.

"The study also lacks clear outcome definition":

This was an exploratory qualitative study of staff experiences of using a relatively new tool in a new way and in new areas of health care. There is little prior research in this area and it would have been inappropriate to pre-empt findings by defining 'clear outcomes' in advance of the research, in the sense in which this is important in many quantitative research designs. We define our objective in the abstract ("The objective of this study was to explore staff experiences of using NEWS in these new settings") and provide more detail in the final sentence of the introduction ("Our focus was on how staff used NEWS, their views on the role of the tool in assessing acute-illness, and its role in the escalation of care").

"Study design: While qualitative interviews are useful, what would show benefit is if the authors used structured interviews or surveys so the responses could be compared against different correspondents. The current study design makes it difficult to synthesize the learnings from the interviews & discourages reproducibility":

We intentionally used semi-structured qualitative interviews and open questions to allow participants to fully describe their experiences and introduce themes which might not have been anticipated by our review of existing literature and input to the topic guide from our multi-professional team. Structured interviews or surveys would have imposed greater limitations on the nature of participants' contributions. We did not consider this appropriate in an exploratory study in an area with limited prior research. 'Reproducibility' is a quality measure more commonly associated with systematic reviews and quantitative research designs. Given our qualitative study design we aimed to carry out research which was reliable, credible, reflexive and transparent. We believe that we have achieved this, and have discussed the limitations of our approach in the third paragraph of the Discussion section as well as the 'Strengths and limitations' section.

"Due to the qualitative nature of the study the results are presented thematically using direct quotes. While this is interesting to read, it is difficult to grasp the key ideas conveyed by the respondents. It might be useful if the authors could analyze the transcripts and classify the positive & negative comments for each theme":

Our participants were experienced professionals who often expressed nuanced views on NEWS, highlighting tensions between benefits and disadvantages, and introduced historical and contextual factors (as is often the case in qualitative interviews). As such, the interviews provided in-depth qualitative data. Classifying their comments into simply positive or negative for each theme would have both distorted the data and greatly reduced its value. We used data driven inductive thematic analysis, which means we searched for, reviewed, defined and named themes strongly linked to the data itself. As part of this process we identified a range of perspectives from different participants on each theme we developed from the data. We have reported these themes, the ranges of perspectives

(and where appropriate, indicated their prevalence), in the relevant sections of the results, and returned to them in the discussion.

“A more detailed description of the customization of NEWS to different outpatient/ non- acute settings might be useful for others looking to reproduce this work”:

While, as discussed above, ‘reproducibility’ was not our aim, we agree that the customization of NEWS in different settings is an interesting topic which warrants further research. In the current paper we have given an indication of the types of adaptations, and reasons for them, reported by our participants.

“information on users level of clinical experience & years of experience using NEWS would be good to know”:

We agree that duration of clinical experience, and experience of NEWS, are relevant contextual factors in individuals’ use of NEWS. These factors were considered in the analysis, and where pertinent have been mentioned in the results – for example under ‘NEWS and communication’ in relation to perceived clinical status gaps; and under ‘Integrating NEWS into clinical practice’ in relation to how routinely different health care professionals used NEWS in their environment, and to mental health staff’s degree of experience in assessing physical health. However we did not find definitive patterns based on experience – with instances of both relatively new and very experienced practitioners reporting either confidence with, or concerns related to the tool. We have avoided providing a detailed demographics table for our participants – for example giving professional role, sector, extent of clinical experience and extent of experience with NEWS: while we agree that this would be of interest, given the geographically limited nature of the study, providing such details would compromise anonymity. We have added a descriptive paragraph after the current table in the results section, giving further detail on participants’ experience, which we hope will be helpful.

“One major limitation of the study is the relatively small sample size & geographic localization. These should be elaborated on”:

While our overall sample size is appropriate for our study design and methods, we have highlighted in the ‘Strengths and limitations’ section that, due to sampling from a wide range of healthcare settings, we were only able to interview a small number of participants from each setting. While we agree that findings in different geographic areas may be different, at the time of this research, the West of England was the only region where NEWS had been rolled out across the healthcare system – and thus the only region where it was possible to carry out a study of this nature. We have amended the ‘Strengths and limitations’ section to highlight this. In the context of very limited research on the use of NEWS outside acute hospital settings, and emerging recommendations from a number of national organisations including NHS England on implementation of NEWS within the NHS, one objective of this research was to provide timely information for those seeking to implement NEWS in other geographical areas. We agree that it would be of interest to carry out research looking at use of NEWS in different geographical areas in the future.

Reviewer 3 comments

Coding differences and how resolved: We have added a sentence to the methods section giving this information. Coding differences were minor, and discussion indicated that they resulted from some initial differences in our understandings of codes/labels, rather than different interpretations of the data coded. We thus resolved these differences by clarifying and agreeing descriptions for the relevant codes.

Use of reference in results section: We agree that references are best not included in results sections. We have thus rephrased in the three sections of the results where we referred to the RCP recommendations, and we have added two sentences to the second paragraph of the discussion.

Page 9 lines 41-47 missing references: Thank you for pointing out this oversight, introduced during editing, which we have now corrected.

Reviewer 4 comments

- 1) In addition to the footnote, we have now added details of ethical approval to the methods section.
- 2) We have added further details regarding the topic guide to the methods section. A scoping review of relevant literature was carried out by JB and used to inform his first draft of the topic guide to ensure we covered themes of importance which had been identified in the literature. The draft topic guide was shared with our multi-professional study team, which included members with experience of using NEWS in a range of contexts, and who provided further input. The topic guide was worded, and interviews were conducted, in such a way as to allow participants to raise topics of importance to them. The topic guide was thereafter modified as data analysis progressed.
- 3) We agree that duration of clinical experience, and experience of NEWS, are relevant contextual factors in individuals' use of NEWS. These factors were considered in the analysis, and where pertinent have been mentioned in the results – for example under 'NEWS and communication' in relation to perceived clinical status gaps; and under 'Integrating NEWS into clinical practice' in relation to how routinely different health care professionals used NEWS in their environment, and to mental health staff's degree of experience in assessing physical health. However we did not find definitive patterns based on experience – with instances of both relatively new and very experienced practitioners reporting either confidence with, or concerns related to the tool. We have avoided providing a detailed demographics table for our participants – for example giving professional role, sector, extent of clinical experience and extent of experience with NEWS: while we agree that this would be of interest, given the geographically limited nature of the study, providing such details would compromise anonymity. We have added a descriptive paragraph after the current table in the results section, giving further detail on participants' experience, which we hope will be helpful..

VERSION 2 – REVIEW

REVIEWER	Erina Ghosh Philips Research North America, US
REVIEW RETURNED	18-Jul-2018
GENERAL COMMENTS	I have no further comments. The authors have addressed my comments.
REVIEWER	Miss Candice Downey University of Leeds, United Kingdom
REVIEW RETURNED	14-Jul-2018
GENERAL COMMENTS	I have no further revisions to suggest. I recommend acceptance for publication.